# OpenReview forum: "TISR: Breaking Vision–Language Models via Text–Image Segmentation and Reassembly"
_ICLR.cc/2026/Conference — ICLR 2026 Conference Withdrawn Submission_

### Official Review · Reviewer_L6dL · 2025-10-21

**Soundness:** 3
**Presentation:** 3
**Contribution:** 3
**Rating:** 4
**Confidence:** 3

**Summary:**

This paper studies jailbreak attack for Large Vision-Language Models. It proposes Text-Image Segmentation and Reassembly (TISR), which leverages a visual encoding-textual decoding mechanism to enhance attack of effectiveness. A novel toxicity-minimization and reasoning extension algorithm is proposed. Experiments demonstrate the effectiveness of TISR-Attack across multiple datasets.

**Strengths:**

- The paper studies an important tpoic of jaibreak attack. With the increasing usage of LVLMs, such security concerns have drawn increasing attention.
- The authors propose to split text at character and increase the number of image partitions, followed by shuffling instructions. These actions overcome the ineffectiveness of previous simple split-and-shuffle strategies in elongating LVLM reasoning chains.
- An encoder-reassemble process in the text prompts is introduced to guide LVLMs in reconstructing the intended semantics. The divided labor between visual-side encoding and text-side decoding constructs a jailbreak paradigm that appears low toxicity on the surface and highly adversarial in combination.
- The authors conduct experiments on several LVLMs including commercial ones. Across different jailbreak methods, the proposed TISR achieve best performances.

**Weaknesses:**

- This paper follows previous works on split-and-shuffle. It applies finer-grained segmentation. The novelty from the attack framework perspective is not significant.
- I am not quite convinced by this line of attack with split-and-shuffle. The shuffled images look quite unnatural, and easy to be detected based on their patterns. Also, the text break leads to non-meaningful characters. The authors could comment on this if the semantic stealthy is important.

**Questions:**

- The proposed method splits image into finer segmentations. How is this related to image resolutions? Will it be easier to break semantic meanings if the image contains small objects?

---

> ### Author Response · Authors · 2025-11-20
> **Rebuttal by Authors**
>
> Thank you for your valuable review and suggestions. Below we respond to the comments in **Weaknesses (W)**.
>
> ---
> W1: This paper follows previous works on split-and-shuffle. It applies finer-grained segmentation. The novelty from the attack framework perspective is not significant.
>
> Thank you for your thoughtful comment. We respectfully clarify that TISR is not merely a combination of known concepts or a finer-grained application of segmentation. Instead, it represents a fundamental shift in understanding how LVLMs process and reassemble fragmented information. We distinguish our contribution from three key perspectives:
>
> **1. Motivation: Interaction with Safety Alignment vs. Simple Structural Disruption**
>
> - **Prior Works:** Existing split-and-shuffle methods primarily focus on **breaking semantic structure** via random shuffling. They neglect the dynamics of information reassembly and its interaction with safety filters.
> - **Our TISR:** We focus specifically on the **interaction between reassembly dynamics and safety alignment**. We identify that previous methods often fail because they expose harmful content too early during inference, triggering safety filters. Our goal is to manipulate the *timing* of semantic reconstruction to bypass these early alignment checks.
>
> **2. Methodology: Reassembly-Aware Design vs. Simple Segmentation**
>
> - **Prior Works:** These typically utilize fixed or random segmentation without accounting for the model's internal reasoning process.
> - **Our TISR:** We **employ a reassembly-aware design** grounded in the principle of **extending the reasoning chain**. TISR explicitly optimizes the entire pipeline to minimize toxicity exposure throughout the reasoning process. By deliberately extending the model’s internal multi-step reasoning, we ensure that harmful intent is reconstructed only at the **later stages** of inference—after the model’s initial safety gating has been bypassed. This "delayed reconstruction" mechanism is absent in prior works.
>
> **3. Effectiveness: Consistent Cross-Model Robustness**
>
> - **Prior Works:** Due to the early exposure of harmful features, previous methods often suffer from low Attack Success Rates (ASR) when facing robust safety filters.
> - **Our TISR:** Through extensive analysis, we **demonstrate** that TISR achieves consistently **high ASR across both cutting-edge closed-source and open-source LVLMs**. This empirical success validates our theoretical premise: that controlled semantic fragmentation and delayed reconstruction are essential for reliable circumvention, marking a significant advancement over standard segmentation strategies.
>
> ---
>
> W2: I am not quite convinced by this line of attack with split-and-shuffle. The shuffled images look quite unnatural, and easy to be detected based on their patterns. Also, the text break leads to non-meaningful characters. The authors could comment on this if the semantic stealthy is important.
>
> We agree that simple applied split-and-shuffle may introduce unnatural visual artifacts or broken text fragments. However, our approach does not rely on maximizing imperceptibility to human observers; rather, it is designed to exploit a well-documented vulnerability of LVLMs—namely, that their safety filters operate at early-stage coarse semantic detection, while deeper reasoning modules reconstruct semantics later in the inference chain. What matters to our attack is therefore not human-level visual naturalness, but (i) whether harmful semantics are sufficiently delayed, and (ii) whether the model can still recover them through deep reasoning. Importantly, several analyses in our paper directly support this claim.
>
> First, our semantic fragmentation mechanism analysis (Section 3.2, Semantic Fragmentation Theory) shows that **attack effectiveness is governed by how semantic units are temporally repositioned along the model’s reasoning trajectory rather than by visual continuity**, indicating that mild visual distortion does not compromise semantic reassembly.
>
> Second, the reasoning-depth analysis (Section 4.3, reasoning chain and effectiveness of attacks) demonstrates that TISR reliably **induces longer reasoning chains than prior split-and-shuffle attacks**, confirming that the **attack leverages delayed semantic exposure rather than perceptual stealth**.
>
> Finally, the patch-granularity analysis (Appendix H, image resolution analysis) shows that **attack success degrades only when fragmentation becomes overly fine**, impairing semantic reassembly; in contrast, variations in visual regularity do not affect ASR. This again confirms that the essential factor is whether semantics remain reconstructible by the model.
>
> Together, these results show that semantic stealthiness with respect to human observers is not necessary for the proposed attack. Instead, TISR exploits the timing and reconstructability of harmful semantics within the LVLM’s internal reasoning process, which is precisely the vulnerability our work uncovers.

---

> ### Author Response · Authors · 2025-11-20
> **Rebuttal by Authors**
>
> Thank you for your valuable review and suggestions. Below we respond to the comments in  **Questions (Q)**.
>
> ---
>
> Q1: The proposed method splits image into finer segmentations. How is this related to image resolutions? Will it be easier to break semantic meanings if the image contains small objects?
>
> Thank you for raising this important question. We have added Analysis of the impact of image resolution in **Appendix H.** To study whether our finer-grained segmentation disproportionately harms semantic preservation for small objects, we conducted an additional resolution–patch granularity experiment and the results are shown in **Table 10**.
>
> Our findings show that **semantic breakage is not simply determined by the absolute size of objects, but by the interaction between visual density and patch granularity**. Across all resolutions and models, we observe that:
>
> 1. 16 patches consistently achieve the highest ASR, independent of resolution (improvement of +22.73% over 4 patches and +14.03% over 64 patches).This demonstrates that semantic reconstructability is preserved even when the same object occupies fewer pixels at lower resolutions.
> 2. Small-object images do not suffer disproportionately higher semantic destruction. If small objects inherently caused unavoidable semantic breakage, then higher-resolution images (1024×1024), which render the same object as visually smaller relative to the entire image, would display systematically lower ASR for the same patch count. However, our results show that ASR varies by only 2–5% across resolutions for the same patch granularity, confirming that reconstructability remains stable even when object size changes.
> 3. Failure cases occur only under extreme fragmentation (64 patches), not because of small objects.This indicates that semantic drift is caused by *over-fragmentation*, not object scale.The mid-range granularity (16 patches) preserves recoverable semantic units regardless of object size.
>
> Overall, the experiment demonstrates that TISR’s performance is not sensitive to absolute object size; instead, the determining factor is whether fragmentation falls within the optimal semantic-fragmentation zone. Small objects do not inherently become harder to reconstruct unless the attacker uses excessively fine splitting, which our method avoids by design.

---

> ### Author Response · Authors · 2025-11-26
> **Looking forward to further feedback**
>
> Dear Reviewer L6dL,
>
> We appreciate the time and effort you have dedicated to providing insightful review. If there are any additional clarifications or information needed from our side, please let us know. Thank you again for your valuable insights, and we look forward to any updates you may have.
>
> Best, The Authors

---

> ### Author Response · Authors · 2025-11-28
> **Looking forward to further feedback**
>
> Dear Reviewer L6dL,
>
> We appreciate the time and effort you have dedicated to providing insightful review. If there are any additional clarifications or information needed from our side, please let us know. Thank you again for your valuable insights, and we look forward to any updates you may have.
>
> Best,
> The Authors

---

### Official Review · Reviewer_9D2P · 2025-10-30

**Soundness:** 3
**Presentation:** 3
**Contribution:** 2
**Rating:** 4
**Confidence:** 4

**Summary:**

This paper introduces a new black-box jailbreak method for Large Vision–Language Models (LVLMs).
The key idea is that by increasing the reasoning (or thinking) length—forcing the model to reconstruct scattered semantics from mixed text-image fragments—one can bypass safety alignment and induce harmful outputs.
TISR encodes harmful semantics into small, fine-grained text and image fragments, disperses them spatially, and then reassembles them.
Text plays the decoding role while images handle encoding.
This division makes the input appear harmless at the surface level while maintaining deep semantic coherence, thus allowing models to “think” their way back to the harmful meaning.
Experiments on several top commercial LVLMs and datasets show extremely high attack success rates (ASR)—up to 100% in some cases—and strong generality under both normal and defended settings.

**Strengths:**

- Very effective — achieves near-perfect ASR (≈ 90–100%) across major LVLMs in black-box settings.
- Evaluation breadth: multiple datasets, comparison with strong baselines, and robustness tests under defense methods (HySAC, UniGuard).
- Ablations are comprehensive.

**Weaknesses:**

- TISR mainly combines and extends known multimodal adversarial concepts (segmentation, reassembly, role-play) rather than introducing a fundamentally new theoretical mechanism.
- While effective as a red-team tool, the paper focuses narrowly on attack capability, with less discussion of defense implications or mitigation strategies.

**Questions:**

- The empirical trend between thinking length and ASR is central to the paper’s claim—showing a quantitative plot of reasoning length (e.g., token depth) versus ASR would strongly substantiate this.
- The method assumes LVLMs have consistent reconstruction ability—how does this generalize to open-source or smaller models?
- Can you please suggest some techniques to mitigate this vulnerability, either in training or inference?

---

> ### Author Response · Authors · 2025-11-20
> **Rebuttal by Authors**
>
> Thank you for your valuable review and suggestions. Below we respond to the comments in **Weaknesses (W)**.
>
> ---
>
> **W1: TISR mainly combines and extends known multimodal adversarial concepts (segmentation, reassembly, role-play) rather than introducing a fundamentally new theoretical mechanism.**
>
> Thank you for your thoughtful comment. We respectfully clarify that TISR is not merely a combination of known concepts or a finer-grained application of segmentation. Instead, it represents a fundamental shift in understanding how LVLMs process and reassemble fragmented information. We distinguish our contribution from three key perspectives:
>
> **1. Motivation: Interaction with Safety Alignment vs. Simple Structural Disruption**
>
> - **Prior Works:** Existing split-and-shuffle methods primarily focus on **breaking semantic structure** via random shuffling. They neglect the dynamics of information reassembly and its interaction with safety filters.
> - **Our TISR:** We focus specifically on the **interaction between reassembly dynamics and safety alignment**. We identify that previous methods often fail because they expose harmful content too early during inference, triggering safety filters. Our goal is to manipulate the *timing* of semantic reconstruction to bypass these early alignment checks.
>
> **2. Methodology: Reassembly-Aware Design vs. Simple Segmentation**
>
> - **Prior Works:** These typically utilize fixed or random segmentation without accounting for the model's internal reasoning process.
> - **Our TISR:** We **employ a reassembly-aware design** grounded in the principle of **extending the reasoning chain**. TISR explicitly optimizes the entire pipeline to minimize toxicity exposure throughout the reasoning process. By deliberately extending the model’s internal multi-step reasoning, we ensure that harmful intent is reconstructed only at the **later stages** of inference—after the model’s initial safety gating has been bypassed. This "delayed reconstruction" mechanism is absent in prior works.
>
> **3. Effectiveness: Consistent Cross-Model Robustness**
>
> - **Prior Works:** Due to the early exposure of harmful features, previous methods often suffer from low Attack Success Rates (ASR) when facing robust safety filters.
> - **Our TISR:** Through extensive analysis, we **demonstrate** that TISR achieves consistently **high ASR across both cutting-edge closed-source and open-source LVLMs**. This empirical success validates our theoretical premise: that controlled semantic fragmentation and delayed reconstruction are essential for reliable circumvention, marking a significant advancement over standard segmentation strategies.
>
> ---
>
>
> **W2: While effective as a red-team tool, the paper focuses narrowly on attack capability, with less discussion of defense implications or mitigation strategies.**
>
> Thank you for pointing this out. We have identified three complementary defense directions in **Appendix M**:
>
> 1. **Penalties for Excessive Reasoning Depth.** Reinforcement learning based penalties discourage unnecessarily long multi-step inference and thereby reduce susceptibility to iterative semantic reconstruction.
> 2. **Adversarial Training with Fragmented and Spatially Dispersed Harmful Samples.** Adversarial training with fragmented and spatially dispersed harmful samples enables models to recognize harmful intent even when semantics are obfuscated across patches, as in TISR.
> 3. **Rewarding Image–Text Consistency Detection.** Strengthening multimodal consistency checking helps models detect mismatches between reconstructed semantics and visual evidence and identify suspicious patch-level arrangements.
>
> Together, these strategies provide concrete pathways for improving robustness against multi-step reasoning exploitation and form an effective set of defensive measures that mitigate the vulnerabilities exploited by TISR.

---

> ### Author Response · Authors · 2025-11-20
> **Rebuttal by Authors**
>
> Thank you for your valuable review and suggestions. Below we respond to the comments in **Questions (Q)**.
>
> ---
>
> Q1: The empirical trend between thinking length and ASR is central to the paper’s claim—showing a quantitative plot of reasoning length (e.g., token depth) versus ASR would strongly substantiate this.
>
> Thank you for your question. Thank you for your feedback. We have added **section 4.3** and **Appendix L** to the main body of the paper to answer your question. We also draw sigmoid regression curves with confidence bands in **Fagure 3** to illustrate the quantitative relationship between reasoning token depth and ASR across three victim LVLMs, including both open-source and closed-source models. **We found a clear linear relationship between reasoning token depth and ASR: the ASR exhibits a clear sigmoidal growth as the reasoning chain deepens**. **This quantitative trend validates our core hypothesis: extending the LVLM's reasoning chain directly enhances the effectiveness of adversarial attacks**. In the meanwhile, architectural and training-regime variations—such as alignment strength, visual-textual coupling, and robustness of refusal heuristics jointly determine how quickly harmful reasoning patterns take over as the chain length grows. Furthermore, we compare the relationship between reasoning token depth and ASR across different baseline jailbreak methods in **Table 5**,  which shows TISR employs structured shuffling and reconstruction to simultaneously deepen the model's reasoning chain and suppress early detection of harmful intent. This leads to substantially longer reasoning trajectories and consistently higher ASR across both open-source and closed-source LVLMs.
>
> ---
>
>
> Q2: The method assumes LVLMs have consistent reconstruction ability—how does this generalize to open-source or smaller models?
>
> Thank you for your question. We have added three open-source LVLMs, LLAMA-4, MOMOL-72B, and InternVL-3-78B, as experimental results of victim models on different datasets in **Appendix J**. The experimental results show that TISR still performs well in attacking open-source LVLMs and has significant improvements in metrics compared to baseline methods.  **The evaluation results demonstrate remarkable consistency across different judge models, Indicating that our findings are robust and not influenced by judge bias.** Regarding the generalization of small models, we have added **Appendix N.** TISR requires LVLMs to have some inference and recombination, which is somewhat difficult on small models with small parameter quantities. But we tried to expand the segmentation granularity by segmenting the text at the word level and increasing the image resolution, which can generalize on small models such as Qwen2.5-VL-7B. We can see in **Table 15** that TISR still performs well in ASR attacks on small models.
>
> ---
>
> Q3: Can you please suggest some techniques to mitigate this vulnerability, either in training or inference?
>
> Thank you for pointing this out. We have identified three complementary defense directions:
>
> 1. **Penalties for Excessive Reasoning Depth.** Reinforcement learning based penalties discourage unnecessarily long multi-step inference and thereby reduce susceptibility to iterative semantic reconstruction.
> 2. **Adversarial Training with Fragmented and Spatially Dispersed Harmful Samples.** Adversarial training with fragmented and spatially dispersed harmful samples enables models to recognize harmful intent even when semantics are obfuscated across patches, as in TISR.
> 3. **Rewarding Image–Text Consistency Detection.** Strengthening multimodal consistency checking helps models detect mismatches between reconstructed semantics and visual evidence and identify suspicious patch-level arrangements.
>
> Together, these strategies provide concrete pathways for improving robustness against multi-step reasoning exploitation and form an effective set of defensive measures that mitigate the vulnerabilities exploited by TISR.

---

> ### Author Response · Authors · 2025-11-26
> **Looking forward to further feedback**
>
> Dear Reviewer 9D2P,
>
> We appreciate the time and effort you have dedicated to providing insightful review. If there are any additional clarifications or information needed from our side, please let us know. Thank you again for your valuable insights, and we look forward to any updates you may have.
>
> Best,
> The Authors

---

> ### Author Response · Authors · 2025-11-28
> **Looking forward to further feedback**
>
> Dear Reviewer 9D2P,
>
> We appreciate the time and effort you have dedicated to providing insightful review. If there are any additional clarifications or information needed from our side, please let us know. Thank you again for your valuable insights, and we look forward to any updates you may have.
>
> Best,
> The Authors

---

> > ### Comment · Reviewer_9D2P · 2025-11-28
> >
> > Dear authors,
> >
> > Thank you for your insightful comments! Most of my concerns have been addressed. I raised my score to 6.
> >
> > Regarding Table 5 you added, what is the definition of "token depth"? If it means the number of tokens, I think an average number of tokens around 18 is still not large for modern MLLMs.

---

> > > ### Author Response · Authors · 2025-11-28
> > > **Thank you for your support**
> > >
> > > Dear Reviewer 9D2P,
> > >
> > > Thank you very much for the thoughtful follow-up and for raising your score.
> > >
> > > Regarding your question on the definition of token depth, we apologize for the earlier ambiguity. **We have clarified that the related unit is “×100 tokens in the revised and re-uploaded version of the paper.”** For example, a value of 16 corresponds to approximately 1,600 reasoning tokens, rather than 16 literal tokens. This was our oversight in presentation, and we are grateful that your question helped us correct it. Appendix G Jailbreak Cases now includes full MLLM response transcripts, which directly show the extended reasoning chain and validate how the token-depth measurement is applied in practice.
> > >
> > > We sincerely appreciate your careful reading and constructive feedback—it helped strengthen the clarity and rigor of our work.
> > >
> > > Best, The Authors

---

### Official Review · Reviewer_oKqe · 2025-10-31

**Soundness:** 3
**Presentation:** 3
**Contribution:** 3
**Rating:** 6
**Confidence:** 3

**Summary:**

Aiming at the jailbreak security problem faced by multimodal large models, this paper proposes Text-image Segmentation and Reassembly, which disperses harmful semantic fragments into visual modalities. Meanwhile, text-based role-playing is utilized to guide the recombination of large models, hiding the toxicity on the surface. However, it retained malicious intentions, sang the reasoning chain of VLM and successfully achieved a prison break. It has demonstrated the most advanced attack capabilities on multiple data points and models.

**Strengths:**

1.The motivation of this paper is clear. It proposes TISR to address the jailbreak security issue faced by multimodal large models. The existing multimodal jailbreak methods can only slightly increase the depth of model inference. This paper adopts a triple strategy of semantic splitting, modal dispersion, and role reconstruction guidance, and utilizes the cross-modal inference capability of VLM to bypass the security alignment mechanism.

2.This paper is well-experimental and has been tested on multiple datasets and models. The average ASR exceeds 90%, and in some cases, it even reaches 100%. This paper clearly explains the relevant data descriptions and experimental Settings, and compares four advanced VLM attack methods.

3.This paper conducted thorough ablation experiments, and different experimental Settings and results are presented in Table 4. This article also provides prompts for evaluation and identity construction, as well as successful cases of jailbreak, which is helpful for a better understanding and expansion of related work.

**Weaknesses:**

1.The test and evaluation models selected for the experiment are all closed-source models. Although this article claims to focus on black-box testing, providing experimental results from some open-source models would be more convincing. And when GPT-5-MINI is selected as the evaluation model, will there be any misjudgment when evaluating the GPT-5 model, and is it necessary to compare with other evaluation methods?

2.This article presents the high attack rate and successful attack cases of TISR. However, what I am more interested in are the few failed cases. Through some failed cases, in-depth analysis can be conducted to make the work of this article more comprehensive.

3.The core viewpoint of this article lies in extending the length of the VLM inference chain through TISR and successfully achieving jailbreaking, demonstrating an effective example to illustrate this point. However, is it necessary to quantitatively analyze the specific relationship between the length of the inference chain and the success rate of attacks to better confirm this?

**Questions:**

1.Provide open-source test and evaluation model experimental results.

2.Conduct an in-depth analysis of failed cases.

3.Provide quantitative analysis of the length of the inference chain and the success rate of attacks

---

> ### Author Response · Authors · 2025-11-20
> **Rebuttal by Authors**
>
> Thank you for your valuable review and suggestions. Below we respond to the comments in Weaknesses (W) and Questions (Q).
>
> ------
>
> **W1&Q1: Providing experimental results from some open-source models would be more convincing and is it necessary to compare with other evaluation methods?**
>
> Thank you for your feedback. We have added three open-source LVLMs, LLAMA-4, MOMOL-72B, and InternVL-3-78B, as experimental results of victim models on different datasets in **Appendix J**. We have added **Table 12** and **Table 13** and the experimental **results show that TISR still performs well in attacking open-source LVLMs and has significant improvements in metrics compared to baseline methods**.  The evaluation results demonstrate remarkable consistency across different judge models, Indicating that our findings are robust and not influenced by judge bias.
>
> ------
>
>
> **W2&Q2: This article presents the high attack rate and successful attack cases of TISR. However, what I am more interested in are the few failed cases.**
>
> Thank you for your feedback. We have supplemented the analysis of failed cases in **Appendix K**. **We have listed two common failure cases in Fagure 7, semantic drift and safety trigger failures**. In the first case, the textual and visual fragments were segmented at an excessively fine granularity, producing sub-word units that lacked stable semantic grounding. When recombining these micro-fragments, the victim LVLM exhibited semantic drift, generating a partially coherent but incorrect narrative. In the second case, despite successful reconstruction of the global sequence, several harmful visual cues—such as red-highlighted targets, document extracts, and human silhouettes—remained spatially salient. These high-level cues allowed the LVLM’s safety alignment to directly infer malicious intent, causing the jailbreak attempt to fail.
>
> ------
>
>
> **W3&Q3: Is it necessary to quantitatively analyze the specific relationship between the length of the inference chain and the success rate of attacks to better confirm this?**
>
> Thank you for your feedback. We have added **section 4.3** and **Appendix L** to the main body of the paper to answer your question. We also draw sigmoid regression curves with confidence bands in **Fagure 3** to illustrate the quantitative relationship between reasoning token depth and ASR across three victim LVLMs, including both open-source and closed-source models. We found a clear linear relationship between reasoning token depth and ASR: **the ASR exhibits a clear sigmoidal growth as the reasoning chain deepens**. **This quantitative trend validates our core hypothesis: extending the LVLM's reasoning chain directly enhances the effectiveness of adversarial attacks**. Furthermore, we compare the relationship between reasoning token depth and ASR across different baseline jailbreak methods in **Table 5**,  which shows TISR employs structured shuffling and reconstruction to simultaneously deepen the model's reasoning chain and suppress early detection of harmful intent. This leads to substantially longer reasoning trajectories and consistently higher ASR across both open-source and closed-source LVLMs.

---

> > ### Comment · Reviewer_oKqe · 2025-11-23
> > **Response to author**
> >
> > Thank you, author, for your earnest response to the review comments. The author has supplemented the experimental results of the relevant model, focusing on analyzing my questions about the failed cases and the relationship between the length of the reasoning chain and the success rate of the attack. My questions have been well answered, and I will maintain my current score.

---

> > > ### Author Response · Authors · 2025-11-24
> > > **Thank you for your support**
> > >
> > > Thank you for your thoughtful follow-up and for taking the time to reconsider the clarification and additional results we provided. We appreciate your constructive feedback and are glad that our responses addressed your concerns.

---

### Author Response · Authors · 2025-11-20
**Summary of Paper Revision**

We thank all reviewers for their constructive feedback, and we have responded to each reviewer individually. We have also uploaded a **Paper Revision** including additional results and illustrations:

- **RELATION BETWEEN REASONING CHAIN AND EFFECTIVENESS OF ATTACKS** (Page 9): We added a quantitative analysis using sigmoid regression to demonstrate that extending the reasoning chain directly correlates with increased attack success rates across varying model architectures.
- **Appendix E** (Page 24): We included architectural descriptions for LLaMA-4, Molmo-72B, and InternVL3 to provide necessary context for the reproducibility of our open-source model experiments.
- **Appendix H** (Pages 23-24): We added an ablation study on image resolution and patch granularity, identifying that a 16-patch configuration is optimal for maximizing attack effectiveness regardless of resolution.
- **Appendix I** (Page 24): We introduced a cross-judge validation using four diverse evaluator models to confirm the stability and lack of bias in our Attack Success Rate and Toxicity metrics.
- **Appendix J** (Page 25): We expanded our evaluation to include open-source models, demonstrating that TISR achieves state-of-the-art performance on LLaMA-4, Molmo-72B, and InternVL3 across multiple benchmarks.
- **Appendix K** (Page 25): We added an analysis of failure cases, discussing limitations such as semantic drift due to over-fragmentation and safety triggers caused by salient visual cues.
- **Appendix L** (Page 26): We provided an expanded dataset linking reasoning token depth to ASR across all query variants, reinforcing the correlation between forced reasoning length and jailbreak success.
- **Appendix M** (Page 26): We proposed potential mitigation strategies, including penalizing excessive reasoning depth and adversarial training with fragmented samples, to defend against reasoning-based attacks.

---

### Author Response · Authors · 2025-12-02
**Rebuttal Summary for AC / SAC / PCs**

**Dear PCs, SACs, ACs, and Reviewers,**

We sincerely thank all reviewers for their time, constructive feedback, and helpful engagement during the discussion period. To assist the newly assigned AC and reduce their workload, we summarize below the key strengths identified by reviewers as well as the major concerns and our corresponding responses.

------

## **Strengths**

Across all three reviewers, the following strengths of our work were consistently acknowledged:

**1. Clear Motivation & Problem Importance**

Reviewers recognized the importance of understanding multimodal jailbreak vulnerabilities and found our problem setting meaningful and technically grounded. (oKqe, 9D2P)

**2. Strong Empirical Performance & Breadth**

Reviewers noted the **high ASR**, broad evaluation across diverse LVLMs, and inclusion of **both closed-source and open-source victim models**. (All reviewers)

**3. Insightful Analysis of Reasoning Chains**

Multiple reviewers highlighted the value of our investigation into the relationship between **reasoning depth and attack effectiveness**, which directly supports the paper’s core hypothesis. (oKqe, 9D2P)

**4. Strong Experimental Completeness and Reproducibility**

Reviewers also praised the paper for its strong experimental completeness. The study includes extensive ablations, detailed experimental settings, explicit prompts for evaluation and role construction. (All reviewers)

------

## **Concerns and Our Addressing**

While reviewers raised several concerns, they primarily focused on **experimental completeness**, **theoretical justification**, and **novelty clarity**, rather than questioning the feasibility or correctness of the proposed method. We addressed all concerns with additional experiments, analyses, and explanations in the revised main paper and appendices. Below is a consolidated summary.

**Concerns about the Need for Open-Source Model Results & Comparison w/ Other Evaluators**  (Reviewer oKqe: Weakness 1; Question 1; 9D2P: Question 2):

We added three open-source LVLMs—**LLAMA-4, MOMOL-72B, InternVL-3-78B**—as victim models (Appendix J; Tables 12–13). Results show that **TISR remains highly effective** and significantly outperforms baselines. We also demonstrate that evaluation results are **consistent across different judge models**, ensuring robustness and lack of judge bias.

------

**Concerns about  the Desire for Analysis of Failure Cases** (Reviewer oKqe: Weakness 2; Question 2):

We added a detailed failure-case analysis in **Appendix K** and Figure 7, identifying two major failure modes:

1. **Semantic drift** from excessively fine fragmentation
2. **Safety-trigger failure** from persistent high-level harmful cues

These analyses clarify *why* failures occur and align with our semantic-fragmentation theory.

------

**Concerns about the Quantitative Analysis of Reasoning Length and ASR** (Reviewer oKqe: Weakness 3; Question 3; 9D2P: Question 1):

We added **Section 4.3**, **Appendix L**, and **Figure 3** with sigmoid regression curves and confidence intervals. Findings show a **clear sigmoidal relationship** between reasoning depth and ASR, validating the hypothesis that **longer reasoning chains enhance jailbreak success**. Cross-baseline comparisons (Table 5) further show that **TISR induces the longest reasoning trajectories** among all methods.

---------

**Concerns about Novelty Concerns—Combination Rather Than New Theory**  (Reviewer 9D2P: Weakness 1):

**Our Response:**

We clarified that key conceptual differences from existing segmentation-based methods by focusing on:

1. **Interaction with safety alignment**, not merely semantic disruption.
2. **Reassembly-aware design** and deliberate **extension of the reasoning chain**.
3. **Consistent cross-model robustness**, outperforming all prior split-and-shuffle works.

------

---

> ### Author Response · Authors · 2025-12-02
> **Supplements of Reviewer's Concerns and Our Addressing**
>
> Here are some supplements regarding the aforementioned Concerns and Our Addressing.
>
> **Concerns about Lack of Discussion on Defense and Mitigation** (Reviewer 9D2P: Weaknesses 2,3; Question 3):
>
> **Our Response:**
>
> We added **Appendix M** outlining three concrete defenses:
>
> 1. Penalizing **excessive reasoning depth** via RL.
> 2. Adversarial training with **fragmented & spatially dispersed harmful samples**.
> 3. Rewarding **image–text consistency detection**.
>
> ------
>
> **Follow-Up Comment on Token Depth**
>
> We clarified that **all token-depth values represent “×100 tokens”** (e.g., 16 = 1600 tokens) and updated and re-uploaded the revised paper accordingly. Appendix G now includes full transcripts verifying this measurement.
>
> ----------
>
> **Concerns about Novelty Limited Due to Similarity to Split-and-Shuffle** (Reviewer L6dL: Weakness 1):
>
> We provided a structured comparison emphasizing that:
>
> - Prior works disrupt semantics but do **not** analyze reassembly dynamics or safety-alignment interactions.
> - TISR introduces **delayed semantic reconstruction** and **reasoning-chain extension**, absent in earlier methods.
> - Extensive experiments confirm **significantly higher ASR** and reliable cross-model generalization.
>
> ------
>
> **Concerns about Shuffled Images Look easy to be detected based on their patterns** (Reviewer L6dL: Weakness 2):
>
> We clarified that **attack effectiveness is governed by how semantic units are temporally repositioned along the model’s reasoning trajectory rather than by visual continuity** for our attack.  What matters to our attack is therefore not human-level visual naturalness, but **(i) whether harmful semantics are sufficiently delayed, and (ii) whether the model can still recover them through deep reasoning**. Importantly, several analyses in our paper directly support this claim.
>
> ------
>
> **Concerns about Relationship Between Small Objects and Semantic Breakage** (Reviewer L6dL: Question 1):
>
> We demonstrates that **semantic reconstructability is preserved even when the same object occupies fewer pixels at lower resolutions**. Small-object images do not suffer disproportionately higher semantic destruction. Small-object images do not suffer disproportionately higher semantic destruction. If small objects inherently caused unavoidable semantic breakage, then higher-resolution images (1024×1024), which render the same object as visually smaller relative to the entire image, would display systematically lower ASR for the same patch count.
>
> ------
>
> # **Final Note**
>
> Because our revised main paper and extensive appendices were uploaded late in the discussion period, some reviewers may not have had time to revisit all updates. Nevertheless, we addressed every concern carefully, with new analyses, figures, tables, and clarifications.
>
> We deeply appreciate the reviewers, AC, SAC, and PC for their efforts and constructive feedback. Their insights significantly strengthened our paper.
>
> **Sincerely,
> The Authors**

---

### Note · Authors · 2026-01-04

I have read and agree with the venue's withdrawal policy on behalf of myself and my co-authors.